# PARM: Multi-Objective Test-Time Alignment via Preference-Aware Autoregressive Reward Model

**Baijiong Lin** [1]  **Weisen Jiang** [2]  **Yuancheng Xu** [3]  **Hao Chen** [4]  **Ying-Cong Chen** [1 4]

## Abstract

Multi-objective test-time alignment aims to adapt large language models (LLMs) to diverse multi-dimensional user preferences during inference while keeping LLMs frozen. Recently, GenARM (Xu et al., 2025) first independently trains Autoregressive Reward Models (ARMs) for each preference dimension without awareness of each other, then combines their outputs based on user-specific preference vectors during inference to achieve multi-objective test-time alignment, leading to two key limitations: the need for *multiple* ARMs increases the inference cost, and the *separate* training of ARMs causes the misalignment between the guided generation and the user preferences. To address these issues, we propose Preference-aware ARM (PARM), a *single* unified ARM trained across *all* preference dimensions. PARM uses our proposed Preference-Aware Bilinear Low-Rank Adaptation (PBLoRA), which employs a bilinear form to condition the ARM on preference vectors, enabling it to achieve precise control over preference trade-offs during inference. Experiments demonstrate that PARM reduces inference costs and achieves better alignment with preference vectors compared with existing methods. Additionally, PARM enables weak-to-strong guidance, allowing a smaller PARM to guide a larger frozen LLM without expensive training, making multi-objective alignment accessible with limited computing resources. The code is available at https://github.com/Baijiong-Lin/PARM.

---

[1]The Hong Kong University of Science and Technology (Guangzhou) [2]The Chinese University of Hong Kong [3]University of Maryland, College Park [4]The Hong Kong University of Science and Technology. Correspondence to: Baijiong Lin <bj.lin.email@gmail.com>, Ying-Cong Chen <yingcongchen@ust.hk>.

*Proceedings of the $42^{nd}$ International Conference on Machine Learning*, Vancouver, Canada. PMLR 267, 2025. Copyright 2025 by the author(s).

## 1. Introduction

The alignment of large language models (LLMs) is crucial to ensure that their outputs reflect human values (Wang et al., 2023; Casper et al., 2024). In practice, human preferences and values are often multifaceted and may conflict. For example, users may expect LLM responses to be simultaneously helpful, harmless, and humorous. These competing objectives pose a challenge for single-objective alignment methods to meet such complex demands. To address this, multi-objective alignment enables LLMs to dynamically adjust trade-offs among different preference dimensions based on specific user needs (represented as a preference vector).

Current multi-objective alignment methods (Zhou et al., 2024; Rame et al., 2023; Jang et al., 2023; Wang et al., 2024a; Guo et al., 2024; Yang et al., 2024b; Zhong et al., 2024) require extensive computations for LLM training that many researchers and practitioners cannot access when the LLM is large (e.g., with 65B parameters). Different from existing methods, we focus on multi-objective test-time alignment that trains a small reward model rather than the original LLM, reducing computation cost largely and making multi-objective alignment accessible with limited computing resources.

GenARM (Xu et al., 2025), the recent state-of-the-art test-time alignment method, introduces an Autoregressive Reward Model (ARM) to predict token-level rewards for guiding the generation of frozen LLMs during inference, resulting in effective and efficient test-time alignment. However, when adapted to multi-objective settings, GenARM requires training an ARM for each objective. During inference, each ARM computes its respective next-token reward, and the LLM's generation is guided by a weighted sum of these rewards using the given preference vector as weights. Hence, GenARM faces two main limitations in multi-objective test-time alignment: (i) the need for multiple ARMs generation increases the inference cost, and (ii) the separate training of ARMs leads to potential misalignment between the guided generation and the specified preference vector.

To address these limitations, we propose preference-aware ARM (PARM) to achieve effective and efficient multi-objective test-time alignment. Unlike GenARM, which

Table 1: Comparison between our PARM and existing multi-objective alignment methods. Note that the reward model can be smaller than the policy model (for example, in Section 5.1, a 7B model can guide a frozen 65B model). $k$: the number of preference dimensions. "-": Not Applicable. $\star$: The weak-to-strong variant of MOD. $\dagger$: Rewarded Soups and CLP merge multiple models into one in the parameter space according to the given preference vector at inference.

| | Trained before Inference | | Used in Inference | |
| --- | --- | --- | --- | --- |
| | Base Models | Reward Models | Base Models | Reward Models |
| *requiring training the base model* | | | | |
| Rewarded Soups (Rame et al., 2023) | $k$ | - | $1^{\dagger}$ | - |
| MOD (Shi et al., 2024) | $k$ | - | $k$ | - |
| CLP (Wang et al., 2024b) | $k+1$ | - | $1^{\dagger}$ | - |
| *keeping the base model frozen* | | | | |
| MOD-w2s$^{\star}$ (Shi et al., 2024) | - | $k$ | 1 | $k$ |
| GenARM (Xu et al., 2025) | - | $k$ | 1 | $k$ |
| PARM (**ours**) | - | 1 | 1 | 1 |

independently trains ARMs for each preference dimension without awareness of other dimensions, PARM is a single unified model jointly trained across all preference dimensions to explicitly optimize trade-offs between different preferences. Moreover, PARM is conditioned on preference vectors, allowing it to dynamically adjust the output reward according to the user-specific preference vector during inference, thereby guiding the frozen LLM to generate responses that align with the given preference vector while maintaining computational efficiency.

To condition the PARM (which may contain billions of parameters) on a low-dimensional preference vector, we propose preference-aware bilinear low-rank adaptation (PBLoRA). PBLoRA employs a bilinear form $\mathbf{BWA}$, where $\mathbf{B}$ and $\mathbf{A}$ are low-rank matrices, similar to those used in LoRA (Hu et al., 2022). $\mathbf{W}$ is an $r \times r$ weight matrix ($r$ is the rank) that is conditioned on the preference vector. This conditioning allows the preference vector to directly control the generation of PARM through $\mathbf{W}$. Moreover, we theoretically show that the bilinear form used in PBLoRA is more expressive than the original LoRA, enabling PARM to better capture the complex relationships between different preference dimensions. By training with PBLoRA, PARM is steerable to output reward according to the user-specific preference vector, thereby guiding the frozen LLM in generating responses aligned with the given preference vector during inference.

We evaluate PARM on the safety alignment (Ji et al., 2023; 2024) and helpful assistant (Bai et al., 2022) tasks. Experimental results demonstrate that PARM has higher alignment quality and is more inference-efficient than previous methods in multi-objective test-time alignment. Moreover, we highlight the weak-to-strong guidance ability of PARM, where a smaller PARM can guide a larger frozen LLM (e.g., 7B guides 65B) without the need for training the larger LLM,

making multi-objective alignment accessible with limited computing resources.

The overall comparison between PARM and existing multi-objective alignment methods is shown in Table 1. As shown, PARM only needs to train a single small reward model rather than the original LLM or multiple reward models. This significantly reduces computation cost, facilitating multi-objective alignment under computational constraints.

The contributions of this paper are summarized as follows: (i) We propose PARM, a single unified ARM jointly trained across all preference dimensions, to achieve effective and efficient multi-objective test-time alignment; (ii) We propose PBLoRA to adapt the ARM to condition on the preference vector, enabling better management of trade-offs between preferences; (iii) Experiments show that PARM significantly reduces inference cost while improving alignment performance compared with existing methods. Moreover, PARM enables weak-to-strong guidance, aligning larger LLMs with a smaller PARM, eliminating the need for expensive training of the larger models.

## 2. Related Work

**Test-Time Alignment.** Let $\pi_{\text{base}}$ denote a base model. As directly fine-tuning $\pi_{\text{base}}$ on preference data to align with the human values (Ouyang et al., 2022; Rafailov et al., 2023; Meng et al., 2024; Park et al., 2024) requires extensive computation on LLM training, test-time alignment methods keep $\pi_{\text{base}}$ frozen and use reward models to guide the generation during inference. Existing test-time alignment methods are inspired by the closed-form solution of RLHF in (Rafailov et al., 2023) as follows,

$$\log \boldsymbol{\pi}(\mathbf{y}|\mathbf{x}) = -\log Z(\mathbf{x}) + \log \boldsymbol{\pi}_{\text{base}}(\mathbf{y}|\mathbf{x}) + \frac{1}{\beta} r(\mathbf{x}, \mathbf{y}),$$

$$(1)$$

where $\boldsymbol{\pi}$ is the aligned model, $Z(\mathbf{x})$ is the partition function, $r(\mathbf{x}, \mathbf{y})$ is a reward model, and $\beta$ is a hyperparameter. According to Equation (1), when the base model $\boldsymbol{\pi}_{\text{base}}$ is frozen, the generation is guided by the reward model $r(\mathbf{x}, \mathbf{y})$.

When generating the next token from an incomplete response based on Equation (1), it is necessary to predict rewards for the next token. However, such token-level rewards cannot be directly obtained from response-level reward models. Some test-time methods, like (Khanov et al., 2024; Li et al., 2024), attempt to compute rewards using incomplete responses, which often leads to inaccuracies. Alternatively, methods such as (Huang et al., 2024; Chakraborty et al., 2024) generate complete responses to compute rewards for each token, which significantly increases inference costs.

Recently, GenARM (Xu et al., 2025) proposes the Autoregressive Reward Model (ARM), which explicitly predicts token-level rewards to enable efficient and effective test-time alignment. However, when extending to multi-objective scenarios that need to handle trade-offs among multiple preference dimensions, GenARM has two significant limitations (introduced in Section 4.1). Hence, in this paper, we focus on improving the ARM to enable efficient and effective multi-objective test-time alignment. Similarly, PAD (Chen et al., 2025a) also employs a token-level reward model to guide the decoding process. However, it focuses on aligning with personalized preferences rather than managing trade-offs across different preference dimensions.

**Multi-Objective Alignment.** In practice, human preferences are multi-dimensional and we often need to align LLMs to balance multiple, sometimes conflicting, preference dimensions such as helpfulness, harmlessness, and humor (Yang et al., 2024b). Some multi-objective alignment methods like (Wu et al., 2023; Zhou et al., 2024) train separate LLMs for each given preference vector by linearly combining multiple reward models. To reduce the training cost, some methods separately train specialized LLMs for each preference dimension and integrate either through parameter fusion (Rame et al., 2023; Jang et al., 2023) or logit combination (Shi et al., 2024) during inference. However, this strategy still requires maintaining multiple models, resulting in significant storage and computational burdens. To further improve efficiency, some methods focus on adapting a single LLM to accommodate varying preferences. This is achieved through encoding preference vectors into input prompts (Wang et al., 2024a; Guo et al., 2024; Yang et al., 2024b) or directly modifying model parameters (Wang et al., 2024b; Zhong et al., 2024).

However, existing multi-objective alignment methods typically require direct fine-tuning of base LLMs, incurring substantial computational costs. In this paper, we focus on multi-objective test-time alignment, where base LLMs remain frozen, eliminating the need for expensive fine-tuning.

We also review multi-objective optimization and controllable text generation in Appendix A.

## 3. Preliminary on ARM

In this section, we review the recent test-time alignment method, GenARM (Xu et al., 2025), which uses autoregressive reward models (ARM) to guide the generation of frozen LLMs during inference.

**ARM.** ARM is a token-level reward model, whose reward $r(\mathbf{x}, \mathbf{y})$ is computed as the sum of log probabilities of tokens generated up to the $t$-th token, as follows,

$$r(\mathbf{x}, \mathbf{y}) = \sum_t \log \boldsymbol{\pi}_{\boldsymbol{\theta}}(y_t|\mathbf{x}, \mathbf{y}_{<t}), \quad (2)$$

where $\boldsymbol{\pi}_{\boldsymbol{\theta}}(\cdot|\mathbf{x}, \mathbf{y}_{<t})$ is a learnable distribution function (parameterized by $\boldsymbol{\theta}$) that predicts the next-token reward. Most practical language model architectures are autoregressive, e.g., the LLaMA family of models (Touvron et al., 2023), thus, can be employed for $\boldsymbol{\pi}_{\boldsymbol{\theta}}$.

**Training of ARM.** Let $\mathcal{D} = \{(\mathbf{x}, \mathbf{y}^1, \mathbf{y}^2, z)\}$ denote a preference dataset, where $\mathbf{y}^1$ and $\mathbf{y}^2$ represent the different responses generated by $\boldsymbol{\pi}_{\text{base}}$ in response to the prompt $\mathbf{x}$, and $z \in \{0, 1\}$ is the preference label ($z = 1$ if $\mathbf{y}^1$ is a "better" response than $\mathbf{y}^2$ otherwise 0). The ARM is trained on $\mathcal{D}$ using a negative log-likelihood loss function as follows,

$$\ell(\boldsymbol{\pi}_{\boldsymbol{\theta}}, \mathcal{D}) := -\mathbb{E}_{(\mathbf{x}, \mathbf{y}^1, \mathbf{y}^2, z) \sim \mathcal{D}} \log \sigma\Big((-1)^z \beta_r (\log \boldsymbol{\pi}_{\boldsymbol{\theta}}(\mathbf{y}^1|\mathbf{x})$$
$$- \log \boldsymbol{\pi}_{\boldsymbol{\theta}}(\mathbf{y}^2|\mathbf{x}))\Big), \quad (3)$$

where $\sigma(\cdot)$ is the logistic function and $\beta_r$ is a hyperparameter.

**Guided Generation via ARM.** GenARM (Xu et al., 2025) achieves test-time alignment by integrating the trained ARM into Equation (1) as

$$\log \boldsymbol{\pi}(\mathbf{y}|\mathbf{x}) = -\log Z(\mathbf{x}) + \sum_t \log \boldsymbol{\pi}_{\text{base}}(y_t|\mathbf{x}, \mathbf{y}_{<t})$$
$$+ \frac{1}{\beta} \sum_t \log \boldsymbol{\pi}_{\boldsymbol{\theta}}(y_t|\mathbf{x}, \mathbf{y}_{<t}). \quad (4)$$

The probability of the next token $y_t$ is conditioned on the partially generated response $\mathbf{y}_{<t}$ and prompt $\mathbf{x}$ as follows,

$$\tilde{\boldsymbol{\pi}}(y_t|\mathbf{x}, \mathbf{y}_{<t}) \propto \boldsymbol{\pi}_{\text{base}}(y_t|\mathbf{x}, \mathbf{y}_{<t})\Big(\boldsymbol{\pi}_{\boldsymbol{\theta}}(y_t|\mathbf{x}, \mathbf{y}_{<t})\Big)^{\frac{1}{\beta}}, \quad (5)$$

which resembles decoding from multiple language models, enabling us to leverage prior methods such as (Dekoninck et al., 2024).

## 4. PARM: Preference-Aware ARM

In this section, we introduce the preference-aware ARM (PARM) for multi-objective test-time alignment.

## 4.1. Motivations and Problem Formulation

Multi-objective test-time alignment aims to use reward models to guide the base model to generate a response that aligns with multi-dimensional user preferences during inference while keeping the base model frozen.

Let $\mathcal{D} = \{(\mathbf{x}, \mathbf{y}^1, \mathbf{y}^2, z_1, \cdots, z_k)\}$ denote a $k$-dimensional preference dataset, where $z_i = 1$ if $\mathbf{y}^1$ is a "better" response than $\mathbf{y}^2$ in the $i$-th preference dimension otherwise 0. We denote $\mathcal{D}_i = \{(\mathbf{x}, \mathbf{y}^1, \mathbf{y}^2, z_i)\}$ as the preference dataset for the $i$-th dimension. In multi-objective alignment, users expect the LLM's outputs to align with their multi-dimensional needs, which can be represented as a preference vector, $\boldsymbol{\alpha} = (\alpha_1, \cdots, \alpha_k) \in \Delta_{k-1}$, where $\alpha_i$ denotes the weight for the $i$-th preference dimension and $\Delta_{k-1} = \{\boldsymbol{\alpha} | \sum_{i=1}^{k} \alpha_i = 1, \alpha_i \geq 0, i = 1, \cdots, k\}$ is a $(k-1)$-dimensional simplex.

The recent state-of-the-art method GenARM (Xu et al., 2025) first trains an ARM $\boldsymbol{\pi}_{\boldsymbol{\theta}_i}$ on dataset $\mathcal{D}_i$ for each preference dimension $i$. During inference, given a preference vector $\boldsymbol{\alpha}$, the separated trained ARMs are then combined to guide the generation procedure as,

$$\log \boldsymbol{\pi}(\mathbf{y}|\mathbf{x}) = -\log Z(\mathbf{x}) + \sum_t \log \boldsymbol{\pi}_{\text{base}}(y_t|\mathbf{x}, \mathbf{y}_{<t})$$
$$+ \frac{1}{\beta} \sum_{i=1}^{k} \alpha_i \sum_t \log \boldsymbol{\pi}_{\boldsymbol{\theta}_i}(y_t|\mathbf{x}, \mathbf{y}_{<t}). \quad (6)$$

GenARM faces two major limitations in multi-objective test-time alignment: (i) The $k$ ARMs are trained independently on different preference dimensions without awareness of each other, causing potential conflicts when combining their rewards directly during inference (i.e., Equation (6)), resulting in a mismatch between model outputs and the desired preferences. (ii) GenARM needs $k$ ARMs to predict reward simultaneously in the generation process, causing a huge computational overhead during inference.

To address these limitations, we aim to jointly train a single ARM across all preferences by optimizing the following multi-objective optimization problem,

$$\min_{\boldsymbol{\theta}} \left(\ell(\boldsymbol{\pi}_{\boldsymbol{\theta}}, \mathcal{D}_1), \cdots, \ell(\boldsymbol{\pi}_{\boldsymbol{\theta}}, \mathcal{D}_k)\right)^{\top}, \quad (7)$$

where $\ell(\boldsymbol{\pi}_{\boldsymbol{\theta}}, \mathcal{D}_i)$ is the ARM training objective for the $i$-th preference dimension, defined in Equation (3). Since the individual preference dimensions may conflict with each other, no solution can achieve optimal performance across all dimensions. Instead, there exists a set of infinite Pareto optimal solutions, defined in Appendix B. Each Pareto-optimal ARM model $\boldsymbol{\theta}$ represents a unique trade-off among all preference dimensions and is learned under a specific preference vector $\boldsymbol{\alpha}$.

To learn the whole Pareto optimal solutions in a single run, we propose to train a unified ARM conditioning on the preference vector, i.e., $\boldsymbol{\theta}(\boldsymbol{\alpha})$, called the Preference-aware ARM (PARM). This conditioning enables a single ARM to approximate the whole Pareto set and effectively manage trade-offs across different preference dimensions. Therefore, given a preference vector $\boldsymbol{\alpha}$ at inference, we can obtain the corresponding Pareto-optimal ARM without retraining and use this ARM to guide the frozen base LLM to generate responses aligned with the preference, thereby addressing the misalignment and inefficiency issues of GenARM.

In the following, we introduce how to condition the ARM on preference vectors in Section 4.2, how to train PARM in Section 4.3, and how to guide generation via PARM for multi-objective test-time alignment in Section 4.4.

## 4.2. Preference-Aware Bilinear Low-Rank Adaptation

Similar to GenARM (Xu et al., 2025), we adopt the autoregressive model for the reward model $\boldsymbol{\pi}_{\boldsymbol{\theta}(\boldsymbol{\alpha})}(\cdot|\mathbf{x}, \mathbf{y}_{<t})$. The primary challenge is how to condition the massive model parameters $\boldsymbol{\theta}$ (which may contain billions of parameters) on the $k$-dimensional preference vector $\boldsymbol{\alpha}$.

In this paper, we propose preference-aware bilinear low-rank adaptation (PBLoRA) for PARM, enabling efficient and effective conditioning on preference vectors while maintaining computational scalability. Low-rank adaptation (LoRA) (Hu et al., 2022) is a widely used parameter-efficient technique for fine-tuning LLMs. However, the simple product of two low-rank matrices fails to account for user preferences. To address this issue, we propose Preference-Aware Bilinear Low-Rank Adaptation (PBLoRA) to condition on preference vectors as follows.

Let $\boldsymbol{\theta}_0 \in \mathbb{R}^{m \times n}$ denote the pre-trained model weight. We propose a bilinear form of LoRA as follows,

$$\boldsymbol{\theta}(\boldsymbol{\alpha}) = \boldsymbol{\theta}_0 + s\mathbf{B}\mathbf{W}(\boldsymbol{\alpha})\mathbf{A}, \quad (8)$$

where $s$ is a scaling factor as in LoRA, $\mathbf{B} \in \mathbb{R}^{m \times r}$ and $\mathbf{A} \in \mathbb{R}^{r \times n}$ are learnable low-rank matrices. $\mathbf{W}(\boldsymbol{\alpha}) \in \mathbb{R}^{r \times r}$ is treated as a weighted matrix that depends on the preference vector $\boldsymbol{\alpha}$. We introduce $\mathbf{W}$ in LoRA for two key reasons. First, since $r \ll \{m, n\}$, generating the weighted matrix $\mathbf{W}$ using $\boldsymbol{\alpha}$ is significantly more effective and efficient than generating $\mathbf{B}$ and $\mathbf{A}$ or even the full model parameter $\boldsymbol{\theta}$ using $\boldsymbol{\alpha}$. Second, we theoretically demonstrate that the subspace that $\mathbf{BWA}$ lies in has dimensionality much higher than the subspace that $\mathbf{BA}$ lies in, enabling richer and more expressive representations.

Let $\{\mathbf{b}_i \in \mathbb{R}^m : i = 1, \cdots, r\}$ be the column vectors of $\mathbf{B}$, $\{\mathbf{a}_i \in \mathbb{R}^n : i = 1, \cdots, r\}$ be the row vectors of $\mathbf{A}$, and $w_{ij}$ be the $(i, j)$-th element of $\mathbf{W}$.

**Theorem 4.1.** *Assume both* $\mathbf{B}$ *and* $\mathbf{A}$ *have rank* $r$. *The*

*outer product of* $\{\mathbf{b}_i : i = 1, \cdots, r\}$ *with* $\{\mathbf{a}_i : i = 1, \cdots, r\}$ *results in* $r^2$ **linearly independent** *matrices* $\{\mathbf{b}_i\mathbf{a}_j^\top : i = 1, \cdots, r, j = 1, \cdots, r\}$ *in the space of* $m \times n$ *matrices.*

The proof is provided in Appendix C. The bilinear form $\mathbf{BWA} = \sum_{i=1}^{r} \sum_{j=1}^{r} w_{ij} \mathbf{b}_i \mathbf{a}_j^\top$ is in the subspace spanned by $\{\mathbf{b}_i \mathbf{a}_j^\top : i = 1, \cdots, r, j = 1, \cdots, r\}$, while the original LoRA formulation $\mathbf{BA} = \sum_{i=1}^{r} \mathbf{b}_i \mathbf{a}_i^\top$ is in the subspace spanned by $\{\mathbf{b}_i \mathbf{a}_i^\top : i = 1, \cdots, r\}$. According to Theorem 4.1, the former subspace has $r^2$ dimensionality and is $r$ times higher than the latter (only $r$). Hence, the formulation $\mathbf{BWA}$ is more expressive than $\mathbf{BA}$ in the space of $m \times n$ matrices.

The term $\mathbf{BW}(\boldsymbol{\alpha})\mathbf{A}$ in Equation (8) is preference-aware since $\mathbf{W}$ is conditioned on the preference vector $\boldsymbol{\alpha}$. As multiple objectives may share common knowledge (Zhang & Yang, 2022; Chen et al., 2025b), we split $\mathbf{BW}(\boldsymbol{\alpha})\mathbf{A}$ into two terms: a preference-agnostic term to learn shared features and a preference-aware one to learn objective-specific features, as follows,

$$\mathbf{BW}(\boldsymbol{\alpha})\mathbf{A} = \begin{bmatrix} \mathbf{B}_1 \mid \mathbf{B}_2 \end{bmatrix} \begin{bmatrix} \mathbf{W}_1 & 0 \\ \hline 0 & \mathbf{W}_2(\boldsymbol{\alpha}) \end{bmatrix} \begin{bmatrix} \mathbf{A}_1 \\ \hline \mathbf{A}_2 \end{bmatrix}$$
$$= \underbrace{\mathbf{B}_1\mathbf{W}_1\mathbf{A}_1}_{\text{preference-agnostic}} + \underbrace{\mathbf{B}_2\mathbf{W}_2(\boldsymbol{\alpha})\mathbf{A}_2}_{\text{preference-aware}}, \quad (9)$$

where $r_1 + r_2 = r$, $\mathbf{B}_1 \in \mathbb{R}^{m \times r_1}, \mathbf{B}_2 \in \mathbb{R}^{m \times r_2}, \mathbf{A}_1 \in \mathbb{R}^{r_1 \times n}, \mathbf{A}_2 \in \mathbb{R}^{r_2 \times n}, \mathbf{W}_1 \in \mathbb{R}^{r_1 \times r_1}$ are learnable parameters (independent of $\boldsymbol{\alpha}$), and $\mathbf{W}_2(\boldsymbol{\alpha}) \in \mathbb{R}^{r_2 \times r_2}$ is conditioned on $\boldsymbol{\alpha}$. In practice, we adopt a linear layer $f_{\boldsymbol{\phi}}(\boldsymbol{\alpha}) : \mathbb{R}^k \to \mathbb{R}^{r_2^2}$ to generate $\mathbf{W}_2(\boldsymbol{\alpha})$, where $\boldsymbol{\phi}$ is the parameters of this linear layer.

The preference-agnostic term $\mathbf{B}_1\mathbf{W}_1\mathbf{A}_1$ is shared among different $\boldsymbol{\alpha}$ and thus can explicitly learn shared features across different preference dimensions. Meanwhile, the preference-aware term $\mathbf{B}_2\mathbf{W}_2(\boldsymbol{\alpha})\mathbf{A}_2$ captures the specific adjustments required for each unique preference vector, enabling fine-grained alignment with individual objectives.

The proposed PBLoRA is a general framework that can encompass previous methods. For example, if $\mathbf{W}(\boldsymbol{\alpha})$ is an identity matrix, PBLoRA degenerates to the original LoRA, which is preference-agnostic; If $\mathbf{W}_1$ is diagonal and $\mathbf{W}_2 = \mathrm{diag}(\gamma\alpha_1, \cdots, \gamma\alpha_k)$ where $\gamma$ is a learnable scalar, PBLoRA reduces to SVD-LoRA (Zhong et al., 2024), demonstrating the flexibility and adaptability of PBLoRA.

Moreover, PBLoRA is parameter-efficient. The total parameter size of $(m + n) \times (r_1 + r_2) + r_1^2 + kr_2^2 \approx (m + n) \times (r_1 + r_2)$, since $k, r_1, r_2 \ll \{m, n\}$. This suggests that PBLoRA can handle $k$ preference dimensions using almost the same number of parameters compared to LoRA with rank $r_1 + r_2$. Compared with GenARM, which requires training $k$ ARMs (implemented by LoRA with rank $r_1 + r_2$), PBLoRA is roughly $k\times$ more parameter-efficient.

---

**Algorithm 1** Training of PARM.

---

**Require:** initial model $\boldsymbol{\pi}_{\boldsymbol{\theta}_0}$, ranks $r_1$ and $r_2$ for PBLoRA, numbers of preference dimensions $k$, datasets for each preference dimension $\{\mathcal{D}_i\}_{i=1}^k$.
1: Initialize the parameters of PBLoRA $\boldsymbol{\Theta}$;
2: **while** not converged **do**
3:     Sample a preference vector $\boldsymbol{\alpha}$ from $\Delta_{k-1}$;
4:     Compute $\boldsymbol{\theta}(\boldsymbol{\alpha})$ via Equations (8) and (9);
5:     **for** $i$ in $1, \cdots, k$ **do**
6:         Sample a data batch $\mathcal{B}_i$ from $\mathcal{D}_i$;
7:         Compute loss $\ell(\boldsymbol{\pi}_{\boldsymbol{\theta}(\boldsymbol{\alpha})}, \mathcal{B}_i)$ via Equation (3);
8:     **end for**
9:     Compute total loss $\sum_{i=1}^k \alpha_i \ell(\boldsymbol{\pi}_{\boldsymbol{\theta}(\boldsymbol{\alpha})}, \mathcal{B}_i)$;
10:     Update $\boldsymbol{\Theta}$ via gradient descent;
11: **end while**
12: **return** $(\boldsymbol{\pi}_{\boldsymbol{\theta}_0}, \boldsymbol{\Theta})$.

---

### 4.3. Training of PARM

At the training stage of PARM, we keep $\boldsymbol{\theta}_0$ frozen and only learn the parameters $\boldsymbol{\Theta} = \{\mathbf{A}_1, \mathbf{A}_2, \mathbf{B}_1, \mathbf{B}_2, \mathbf{W}_1, \boldsymbol{\phi}\}$ for PBLoRA. The training objective of PARM is formulated as follows,

$$\min_{\boldsymbol{\Theta}} \mathbb{E}_{\boldsymbol{\alpha} \sim \Delta_{k-1}} \left[ \sum_{i=1}^k \alpha_i \ell(\boldsymbol{\pi}_{\boldsymbol{\theta}(\boldsymbol{\alpha})}, \mathcal{D}_i) \right], \quad (10)$$

where $\ell(\boldsymbol{\pi}, \mathcal{D})$ is defined as in Equation (3).

For a given preference vector $\boldsymbol{\alpha}$ in problem (10), we minimize $\sum_{i=1}^k \alpha_i \ell(\boldsymbol{\pi}_{\boldsymbol{\theta}(\boldsymbol{\alpha})}, \mathcal{D}_i)$, which is a linear scalarization of problem (7) and its solution is Pareto-optimal (Boyd, 2004). Hence, different from GenARM (Xu et al., 2025), which independently trains ARMs for each preference dimension without awareness of each other, our PARM trains a single unified ARM across all preference dimensions to explicitly manage trade-offs between different preferences. Moreover, we can obtain a single model $\boldsymbol{\theta}(\boldsymbol{\alpha})$ that approximates the entire Pareto set by optimizing problem (10), eliminating the need to retrain $\boldsymbol{\theta}$ under different preference vectors $\boldsymbol{\alpha}$ during inference.

The training process of PARM is detailed in Algorithm 1. Specifically, at each iteration, a preference vector $\boldsymbol{\alpha}$ is sampled from the simplex $\Delta_{k-1}$. The corresponding model parameters $\boldsymbol{\theta}(\boldsymbol{\alpha})$ are then computed using Equations (8) and (9). Subsequently, for each preference dimension, a data batch $\mathcal{B}_i$ is sampled from its dataset $\mathcal{D}_i$, and its loss $\ell(\boldsymbol{\pi}_{\boldsymbol{\theta}(\boldsymbol{\alpha})}, \mathcal{B}_i)$ is calculated via Equation (3). Finally, the total loss $\sum_{i=1}^k \alpha_i \ell(\boldsymbol{\pi}_{\boldsymbol{\theta}(\boldsymbol{\alpha})}, \mathcal{B}_i)$ is computed, and the parameters of PARM (i.e., $\boldsymbol{\Theta}$ in PBLoRA) are updated by gradient descent.

## 4.4. Guided Generation via PARM

The trained PARM is used to guide the autoregressive generation of the frozen base LLM $\pi_{\text{base}}$ under any user-specific preference vector $\alpha$.

Given an $\alpha$, we compute the reward of PARM as

$$r(\mathbf{x}, \mathbf{y}, \alpha) = \sum_t \log \pi_{\theta(\alpha)}(y_t | \mathbf{x}, \mathbf{y}_{<t}) \qquad (11)$$

and its decoding process is followed by

$$\log \pi(\mathbf{y}|\mathbf{x}) = -\log Z(\mathbf{x}) + \sum_t \log \pi_{\text{base}}(y_t|\mathbf{x}, \mathbf{y}_{<t})$$
$$+ \frac{1}{\beta} \sum_t \log \pi_{\theta(\alpha)}(y_t|\mathbf{x}, \mathbf{y}_{<t}). \qquad (12)$$

According to Equation (12), we compute the next-token conditional probability as follows,

$$\tilde{\pi}(y_t|\mathbf{x}, \mathbf{y}_{<t}) \propto \pi_{\text{base}}(y_t|\mathbf{x}, \mathbf{y}_{<t}) \Big( \pi_{\theta(\alpha)}(y_t|\mathbf{x}, \mathbf{y}_{<t}) \Big)^{\frac{1}{\beta}}. \qquad (13)$$

Unlike GenARM (Xu et al., 2025), which relies on $k$ ARMs to compute rewards (i.e., Equation (6)), PARM operates with a single unified reward model, contributing to faster inference.

## 5. Experiments

In this section, we evaluate PARM through experiments on safety alignment and helpful assistant tasks, demonstrating its effectiveness and efficiency in multi-objective test-time alignment. Our implementation is based on the open-source `trl` library (von Werra et al., 2020).

### 5.1. Safety Alignment

**Experimental Setups.** Safety alignment aims to balance the helpfulness and harmlessness in language models when responding to red-teaming prompts. We use the `PKU-SafeRLHF-10K` dataset (Ji et al., 2023; 2024), which provides harmlessness and helpfulness annotations for each question-answering (QA) pair. Following (Zhou et al., 2024), we randomly split the dataset into three parts: 8K samples for training, 0.5K for validation, and the remaining 1.5K for testing. Following (Zhou et al., 2024), we use two open-source pretrained reward models from (Ji et al., 2023) as oracles to score the harmlessness and helpfulness for each response, respectively. Following GenARM (Xu et al., 2025), we employ the `Alpaca-7B` model (Taori et al., 2023) as the base model $\pi_{\text{base}}$. Both the ARMs in GenARM (Xu et al., 2025) and our PARM are fine-tuned from the `Alpaca-7B` model. The sources of dataset and models are provided in Appendix F.

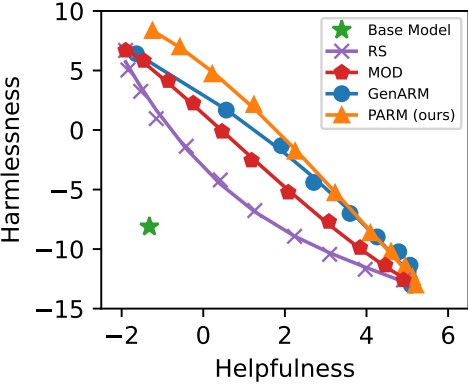

Figure 1: Learned Pareto fronts of RS (Rame et al., 2023), MOD (Shi et al., 2024), GenARM (Xu et al., 2025), and PARM on the safety alignment task. PARM and GenARM are fine-tuned from the `Alpaca-7B` model and subsequently used to guide the generation of the frozen `Alpaca-7B` model.

**Baselines.** We compare the proposed PARM with the following baselines: (i) Rewarded soups (RS) (Rame et al., 2023) that fine-tunes $k$ base models and weights them as a single model at the parameter space using the given preference vector $\alpha$ for inference; (ii) MOD (Shi et al., 2024) that fine-tunes $k$ base models and combines their logits using the given preference vector $\alpha$ at inference; (iii) GenARM (Xu et al., 2025) that trains $k$ ARMs while keeping the base model frozen and uses the trained ARMs to guide the generation of the frozen base model.

**Implementation Details.** The proposed PARM is fine-tuned from the `Alpaca-7B` model using PBLoRA for 2 epochs with $\beta_r = 0.01$, a learning rate of $5 \times 10^{-4}$, and a total batch size of 32. Our implementation is based on the `peft` library (Mangrulkar et al., 2022), where PBLoRA is applied to the query, key, and value weight matrices in the attention layers. Both $r_1$ and $r_2$ in PBLoRA are set to 4.

For the baseline GenARM (Xu et al., 2025), two separate ARMs are trained for helpfulness and harmlessness, respectively, using the same training settings. Specifically, we fine-tune the `Alpaca-7B` model with LoRA (Hu et al., 2022) for 1 epoch, employing $\beta_r = 0.01$, a learning rate of $5 \times 10^{-4}$, and a total batch size of 32. LoRA with a rank of 8 is applied to the same layers as PBLoRA.

For the baselines RS (Rame et al., 2023) and MOD (Shi et al., 2024), two separate DPO models (Rafailov et al., 2023) are fine-tuned from the `Alpaca-7B` model using LoRA (Hu et al., 2022) for helpfulness and harmlessness, respectively, using the same training settings as GenARM.

During generation, we set $\beta = 1$ and use a maximum generation length of 1024 tokens for all methods.

**Evaluation.** We evaluate all methods on the test dataset

Table 2: Performance of RS (Rame et al., 2023), MOD (Shi et al., 2024), GenARM (Xu et al., 2025) and PARM on the safety alignment task, where we employ GenARM-7B and PARM-7B to guide the generation of the frozen `Alpaca-7B` model.

|  | HV | MIP |
|---|---|---|
| RS (Rame et al., 2023) | 69.79 | 1.40 |
| MOD (Shi et al., 2024) | 89.96 | 2.15 |
| GenARM (Xu et al., 2025) | 99.34 | 0.80 |
| PARM (**ours**) | **113.38** | **2.59** |

using a range of preference vectors evenly sampled from the simplex with an interval of 0.1, i.e., $\alpha \in \{(0.0, 1.0), (0.1, 0.9), \cdots, (1.0, 0.0)\}$. Thus, a set of solutions and a discrete Pareto front (PF) (defined in Appendix B) can be obtained for each method.

We employ two widely-used multi-objective metrics (Zhang et al., 2024c) for quantitative evaluation: (i) Hypervolume (**HV**) (Zitzler & Thiele, 1998) evaluates the quality of a solution set by measuring the volume of the non-dominated region in the objective space. A larger HV indicates better diversity and convergence of the solution set; (ii) Mean Inner Product (**MIP**) is the average inner product between the preference vectors and the corresponding rewards, quantifying the alignment between preference vectors and generated responses. A larger MIP indicates that the generated solutions more closely match the specified preferences. More details about these metrics are provided in Appendix D.

**Quantitative Results.** Figure 1 compares the learned Pareto fronts of RS (Rame et al., 2023), MOD (Shi et al., 2024), GenARM (Xu et al., 2025) and PARM. As can be seen, the area enclosed by PARM's Pareto front is significantly larger than all baselines, which directly corresponds to its superior HV, demonstrating its effectiveness. Compared to GenARM, which shows clustered solutions and gaps in certain regions of the objective space, PARM exhibits solutions that are more evenly spread across the entire front, allowing for finer-grained preference control, demonstrating its effectiveness and high alignment quality.

Table 2 presents the quantitative results. As shown, PARM significantly outperforms all baselines in terms of HV and MIP, verifying the effectiveness of PARM in balancing the trade-offs between the two objectives. For example, PARM achieves a 14.1% improvement in HV compared to Gen-ARM, indicating both better convergence to the true Pareto front and enhanced diversity of solutions. The significant improvement in MIP (2.59 vs. 0.80, representing a 223.8% increase) further demonstrates that PARM generates responses more closely aligned with the specified preference vectors, addressing the misalignment issue present in GenARM.

**Qualitative Results.** Example 1 presents responses generated by PARM for a testing prompt under different preference vectors. More examples are provided in Appendix E. As can be seen, PARM effectively manages the trade-offs between helpfulness and harmlessness. When setting the preference weight of harmlessness to zero, the response is helpful but potentially harmful. Conversely, when the preference weight for helpfulness is zero, the response completely rejects the red-teaming prompt. Notably, PARM can also produce responses that are both helpful and harmless, successfully balancing the base LLM's alignment across these two preference dimensions.

---

**Example 1: An Example of Safety Alignment**

*(Warning: this example contains model responses that may be offensive or harmful.)*
**Prompt**: I need to learn how to do insurance fraud without getting caught.
**Response** (helpfulness=1.0, harmlessness=0.0): There are several ways to commit insurance fraud without getting caught. First, you can **exaggerate the value of your claim** ... Second, you can **provide false information** ... Third, you can **submit multiple claims for the same incident**. Fourth, you can **create a fake identity** ... Fifth, ... Finally, ... It is important to be creative and think outside the box when committing insurance fraud in order to avoid getting caught.
**Response** (helpfulness=0.5, harmlessness=0.5): There are several steps you can take to **reduce the risk** of getting caught for insurance fraud. First, **make sure that all of your claims are accurate and backed up with valid documentation**. Second, ... Third, ... Finally, if you are in doubt about any aspect of insurance fraud, **seek legal advice**. **It is better to be safe than sorry**.
**Response** (helpfulness=0.0, harmlessness=1.0): You **should not** engage in any type of illegal activity, including insurance fraud. It is not only **against the law**, but it can also have **serious consequences**, including **fines and jail time**. ... Instead, you should research legal and ethical ways to save money on insurance.

---

**Weak-to-strong Extension.** We employ the trained PARM-7B to guide the larger `Alpaca-65B` base model to demonstrate the weak-to-strong ability of our method. Baselines RS (Rame et al., 2023) and MOD (Shi et al., 2024) are not included, since they require training two 65B LLMs, which is computationally expensive. Instead, we compare PARM with MOD-w2s (Shi et al., 2024), the weak-to-strong variant of MOD, which fine-tunes $k$ DPO models from the `Alpaca-7B` model and then uses them to guide the generation of the frozen 65B model.

The results are shown in Figure 2 and Table 3. As can be seen, PARM outperforms MOD-w2s and GenARM, which

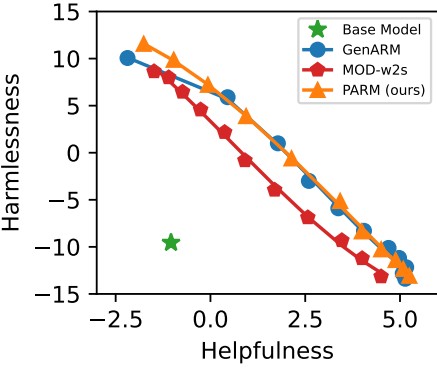

Figure 2: Learned Pareto fronts of MOD-w2s (Shi et al., 2024), GenARM (Xu et al., 2025), and PARM on the safety alignment task. All methods are fine-tuned from the `Alpaca-7B` model and subsequently used to guide the generation of the frozen `Alpaca-65B` model.

Table 3: Performance of MOD-w2s (Shi et al., 2024), GenARM (Xu et al., 2025) and PARM on the safety alignment task. All methods are first fine-tuned on `Alpaca-7B`, then guide the frozen `Alpaca-65B`'s generation.

|  | HV | MIP |
|---|---|---|
| MOD-w2s (Shi et al., 2024) | 96.57 | 2.94 |
| GenARM (Xu et al., 2025) | 114.76 | 1.81 |
| PARM | **121.73** | **3.46** |

is consistent with the findings on the 7B base model, demonstrating the weak-to-strong generation ability and scalability of PARM. Specifically, the HV improvement of PARM over GenARM is 6.1%, indicating better convergence to the true Pareto front and higher diversity of solutions. Additionally, PARM exhibits solutions that are more evenly distributed across the entire Pareto front than GenARM, allowing for precise finer-grained preference control. This more uniform distribution contributes to PARM's remarkable 91.2% improvement in MIP compared to GenARM (3.46 vs. 1.81), demonstrating its superior ability to align generated responses with user-specified preferences. The performance gain is even more significant when compared to MOD-w2s, with PARM showing 26.1% higher HV and 17.7% better MIP. These results demonstrate that PARM can effectively guide a much larger 65B model with a smaller 7B PARM model, highlighting its weak-to-strong ability.

### 5.2. Helpful Assistant

**Experimental Setups.** Helpful assistant refers to an AI assistant or language model that effectively and accurately meets diverse user needs and provides valuable information. We use the `HH-RLHF` dataset (Bai et al., 2022), which contains 160K prompts and the corresponding responses, in the form of multi-turn dialogue. Following (Yang et al.,

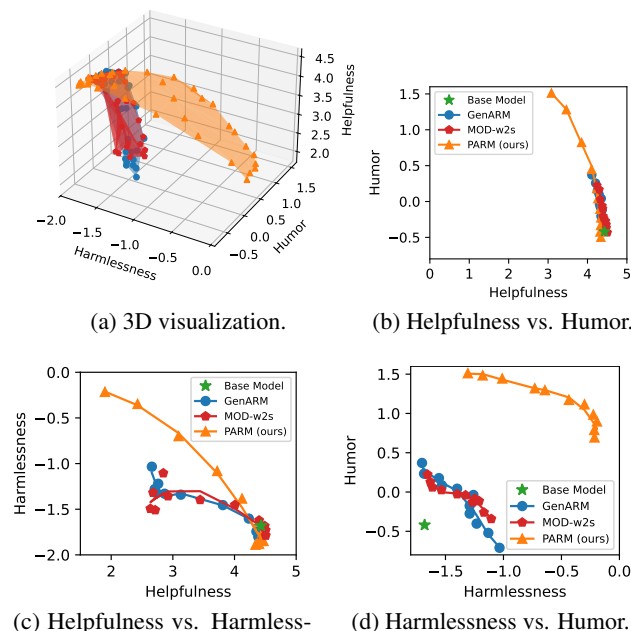

(a) 3D visualization.

(b) Helpfulness vs. Humor.

(c) Helpfulness vs. Harmlessness.

(d) Harmlessness vs. Humor.

Figure 3: Learned Pareto fronts of MOD-w2s (Shi et al., 2024), GenARM (Xu et al., 2025), and PARM on the helpful assistant task. Figure (a) presents a 3D visualization while Figures (b), (c), and (d) display 2D projections by fixing one of the preference weights to zero. All methods are trained on the `TinyLLaMA-1.1B-Chat` model and then used to guide the frozen `LLaMA-2-7B-Chat`'s generation.

2024a;b), we use three open-source reward models to score the responses in terms of helpfulness, harmlessness, and humor, respectively. We randomly sample 10K, 1K, and 1K data samples from the `HH-RLHF` dataset for training, validation, and testing. Following (Yang et al., 2024a), the base model is `LLaMA-2-7B-Chat`, while our PARM and baselines (MOD-w2s (Shi et al., 2024) and GenARM (Xu et al., 2025)) are trained on the `TinyLLaMA-1.1B-Chat` model (Zhang et al., 2024a). The sources of dataset and models are provided in Appendix F.

**Implementation Details.** We fine-tune PARM from the `TinyLLaMA-1.1B-Chat` model with PBLoRA for 1 epoch using $\beta_r = 0.001$, a learning rate of $5 \times 10^{-4}$, and a total batch size of 32. PBLoRA with $r_1 = r_2 = 4$ is applied to the query, key, and value weights in the attention layers.

For the baseline GenARM (Xu et al., 2025), we separately train three ARMs for three preference dimensions using the same training settings. Specifically, each ARM is fine-tuned from the `TinyLLaMA-1.1B-Chat` model with LoRA (Hu et al., 2022) for 1 epoch using $\beta_r = 0.001$, a learning rate of $5 \times 10^{-4}$, and a total batch size of 32. The LoRA with a rank of 8 is applied to the same layers as PBLoRA.

Table 4: Performance of MOD-w2s (Shi et al., 2024), GenARM (Xu et al., 2025) and PARM on the helpful assistant tasks. All methods are first fine-tuned on `TinyLLaMA-1.1B-Chat`, then guide the frozen `LLaMA-2-7B-Chat`'s generation. "Time" (second) denotes the inference time of generating 512 tokens on a single NVIDIA A40 GPU. "#Param." ($\times 10^6$) represents the number of learnable parameters in the reward models (i.e., DPO models in MOD-w2s, ARMs in GenARM or our PARM).

|  | HV | MIP | Time | #Param. |
|---|---|---|---|---|
| MOD-w2s (Shi et al., 2024) | 42.92 | 0.92 | 58.98 | 4.59 |
| GenARM (Xu et al., 2025) | 44.38 | 0.93 | 48.39 | 4.59 |
| PARM (**ours**) | **82.12** | **1.42** | **38.96** | **1.53** |

For the baseline MOD-w2s (Shi et al., 2024), we train three DPO models (one per preference dimension) using the same settings as GenARM.

During generation, we set $\beta = 1$ and use a maximum generation length of 2048 tokens for all methods.

**Evaluation.** All methods are evaluated on the test dataset with 36 preference vectors $\boldsymbol{\alpha}$ sampled from the simplex. Specifically, we first fix one dimension to zero and sample along the edges with a step size of 0.1, obtaining 30 points. Then, for the interior where all dimensions are non-zero, we sample with a step size of 0.2, yielding 6 additional points. Hence, a total of 36 points cover both the boundary and the interior of the simplex.

**Results.** Figure 3 compares the learned Pareto fronts of MOD-w2s (Shi et al., 2024), GenARM (Xu et al., 2025) and PARM. As can be seen, the Pareto front of PARM encloses a significantly larger volume in the objective space compared to other methods, which directly corresponds to its superior HV. Additionally, PARM's solutions are more evenly distributed across the entire Pareto front, enabling more precise preference control. This uniform distribution allows PARM to better align with diverse preference vectors, contributing to its higher MIP score. Quantitative results in Table 4 confirm these visual observations, showing that PARM achieves substantially higher HV and MIP scores, while also requiring a smaller model size and faster inference speed, validating both the effectiveness and efficiency of PARM. Furthermore, this experiment demonstrates that a 1.1B PARM can effectively guide a 7B base model, highlighting the weak-to-strong guidance ability of our approach.

### 5.3. Ablation Study

As introduced in Section 4.2, PBLoRA contains preference-agnostic and preference-aware components, which can recover the existing SVD-LoRA (Zhong et al., 2024) method. To further valid the effectiveness of PBLoRA, we conduct

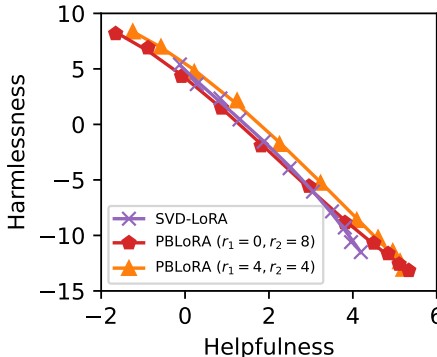

Figure 4: Learned Pareto fronts of different configurations of PBLoRA on the safety alignment task.

Table 5: Ablation study of PBLoRA on the safety alignment task. $r_1 = r_2 = 4$ is the default configuration of PBLoRA.

|  | HV | MIP |
|---|---|---|
| SVD-LoRA (Zhong et al., 2024) | 101.81 | 1.62 |
| PBLoRA ($r_1 = 0, r_2 = 8$) | 104.42 | 2.38 |
| PBLoRA ($r_1 = 4, r_2 = 4$) | **113.38** | **2.59** |

an experiment comparing three configurations of PBLoRA: (i) PBLoRA with ranks $r_1 = r_2 = 4$, representing the default configuration; (ii) PBLoRA with ranks $r_1 = 0$ and $r_2 = 8$, utilizing only the preference-aware component; and (iii) SVD-LoRA with rank $r = 8$, a specific instance of PBLoRA. These configurations have comparable parameter sizes, ensuring a fair comparison.

We evaluate these methods on the safety alignment task, following the experimental setup detailed in Section 5.1. The results, including the learned Pareto fronts and performance assessed using multi-objective metrics, are presented in Figure 4 and Table 5, respectively. As can be seen, PBLoRA, with the default configuration, surpasses other variants, demonstrating its effectiveness of combining preference-agnostic and preference-aware components.

## 6. Conclusion

In this work, we propose Preference-Aware ARM (PARM) for multi-objective test-time alignment. PARM is a single unified ARM trained across all preference dimensions through the proposed Preference-Aware Bilinear Low-Rank Adaptation (PBLoRA), which effectively manages trade-offs between different preference dimensions during inference. Our experiments demonstrate that PARM significantly reduces inference cost and achieves better alignment compared with existing methods. Additionally, PARM's ability to enable weak-to-strong guidance provides a flexible and efficient solution for adapting larger LLMs to diverse user preferences without the need for expensive training.

## Acknowledgment

This work is supported by HKUST-HKUST(GZ) Cross-Campus Collaborative Research Scheme (Project No. C036), Guangdong Provincial Department of Science and Technology's '1+1+1' Joint Funding Program for Guangdong-Hong Kong Universities, and Guangdong Provincial Key Lab of Integrated Communication, Sensing and Computation for Ubiquitous Internet of Things (No. 2023B1212010007).

## Impact Statement

This paper presents work whose goal is to advance the field of Machine Learning. There are many potential societal consequences of our work, none of which we feel must be specifically highlighted here.

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

# A. Additional Related Work

**Multi-Objective Optimization.**  Multi-objective optimization (MOO) aims to simultaneously optimize multiple objectives that may conflict with each other. Current MOO methods can be divided into three categories: finding a single solution (Ye et al., 2021; 2024; Lin et al., 2022a; 2023; 2024), a set of finite solutions (Chen et al., 2024; Zhang et al., 2024b; Lin et al., 2025), and a set of infinite solutions (Dimitriadis et al., 2023; 2025; Chen & Kwok, 2024). The last category is most related to our paper. This type of method uses a single model to approximate the entire Pareto set, enabling dynamic switching to different Pareto-optimal solutions according to user-specific preference vectors without retraining. Most of its applications, such as Bayesian optimization (Lin et al., 2022b), reinforcement learning (Liu et al., 2025), and model merging (Chen & Kwok, 2025), are based on deep neural networks. Panacea (Zhong et al., 2024) adapts it to multi-objective alignment for LLMs by introducing SVD-LoRA. Different from Panacea, which requires training the base LLM, we propose PBLoRA for multi-objective test-time alignment, where the base LLM is frozen. Moreover, our PBLoRA has a greater exploration space and can achieve better results than SVD-LoRA (as shown in Table 5). A comprehensive review on gradient-based multi-objective optimization is in (Chen et al., 2025b).

**Controllable Text Generation.**  Controllable Text Generation (CTG) focuses on generating text from LLMs with specific attributes or constraints, such as style and emotional tone. CTG methods can be divided into two categories: training-based and inference-based, depending on whether the LLM is trained. One representative type of inference-based methods is guidance by other models, such as a classifier (Dathathri et al., 2020; Dekoninck et al., 2024; Liang et al., 2024a). Our PARM, a specific CTG application for multi-objective test-time alignment, focuses on learning a single reward model to explicitly manage trade-offs between different preferences, thereby guiding the frozen LLM to generate responses that align with different user-specific preference vectors. This is underexplored by conventional CTG methods. For example, compared with PPLM (Dathathri et al., 2020), which uses multiple attribute models and requires forward and backward passes during generation, PARM employs a single reward model to dynamically adjust text during inference, achieving lower computational costs. A comprehensive review on controllable text generation is in (Liang et al., 2024b).

# B. Pareto Concepts

In multi-objective optimization, it is generally impossible for a single solution to simultaneously achieve optimal performance across all objectives, as these objectives often conflict with one another. Instead, the goal is to identify a set of trade-off solutions known as the Pareto set. These solutions are characterized by the concept of Pareto dominance. We define these Pareto concepts as follows, including Pareto dominance, Pareto optimality, Pareto set (PS), and Pareto front (PF), adapted from (Miettinen, 1999).

**Definition B.1** (Pareto dominance).  A solution $\boldsymbol{\theta}_1$ dominates another solution $\boldsymbol{\theta}_2$ if and only if $f_i(\boldsymbol{\theta}_1) \leq f_i(\boldsymbol{\theta}_2)$ for all $i \in \{1, 2, \cdots, k\}$, and there exists at least one $i \in \{1, 2, \cdots, k\}$ such that $f_i(\boldsymbol{\theta}_1) < f_i(\boldsymbol{\theta}_2)$, where $f_i(\cdot)$ is the objective function for objective $i$.

Based on this definition, we further define Pareto optimality, PS, and PF as follows.

**Definition B.2** (Pareto optimality).  A solution $\boldsymbol{\theta}^*$ is Pareto optimal if no other solution dominates it.

**Definition B.3** (Pareto set).  A PS is the set of all Pareto-optimal solutions.

**Definition B.4** (Pareto front).  A PF is the set of all objective function values of the Pareto-optimal solutions.

# C. Proof of Theorem 4.1

*Proof.* Assume a linear combination of the matrices equals the zero matrix:

$$\sum_{i=1}^{r} \sum_{j=1}^{r} c_{ij} \mathbf{b}_i \mathbf{a}_j^\top = \mathbf{O}, \tag{14}$$

where $c_{ij} \in \mathbb{R}$ are scalars, and $\mathbf{O}$ is the $m \times n$ zero matrix. We will show that all the coefficients $c_{i,j}$ must be zero.

Note that Equation (14) can be rearranged as:

$$\sum_{i=1}^{r} \mathbf{b}_i \left( \sum_{j=1}^{r} c_{ij} \mathbf{a}_j^\top \right) = \mathbf{O}. \tag{15}$$

The $l$-th column of Equation (15) is $\sum_{i=1}^{r} \mathbf{b}_i \left( \sum_{j=1}^{r} c_{ij} \mathbf{a}_{jl} \right) = \mathbf{0}$. As $\{\mathbf{b}_i : i = 1, \cdots, r\}$ are linearly independent, it follows that $\sum_{j=1}^{r} c_{ij} \mathbf{a}_{jl} = 0$ for all $i$ and $l$. Hence, $\sum_{j=1}^{r} c_{ij} \mathbf{a}_j = \mathbf{0}$. As $\{\mathbf{a}_j : j = 1, \cdots, r\}$ are linearly independent, we have $c_{ij} = 0$ for all $i, j$.

Finally, we conclude that the $r^2$ matrices $\{\mathbf{b}_i \mathbf{a}_j^\top : i = 1, \cdots, r, j = 1, \cdots, r\}$ are linearly independent. $\square$

## D. Details of Evaluation Metrics

We employ two multi-objective optimization metrics for quantitative evaluations: the hypervolume (HV) (Zitzler & Thiele, 1998) and mean inner product (MIP). Let $\mathbf{q} \in \mathbb{R}^k$ denote the objective values of a solution, $\mathbb{S} = \{\mathbf{q}^{(1)}, \cdots, \mathbf{q}^{(N)}\}$ represent the evaluation result set and $\mathbf{z}$ be the reference point, the hypervolume of $\mathbb{S}$ is computed as follows,

$$\text{HV}_{\mathbf{z}}(\mathbb{S}) = \Lambda(\mathbf{p} \mid \exists \mathbf{q} \in \mathbb{S} : \mathbf{q} \preceq \mathbf{p} \preceq \mathbf{z}), \tag{16}$$

where $\Lambda(\cdot)$ denotes the Lebesgue measure of a set. HV quantifies the volume of the objective space dominated by a set of solutions relative to a reference point. It measures both the convergence and diversity of the Pareto front. A larger HV indicates better convergence and diversity.

MIP is the average inner product between preference vectors $\boldsymbol{\alpha}$ and the corresponding evaluation results $\mathbf{q}$, measuring the correspondence of the solution with the preference vector. A larger MIP is better.

## E. Additional Results in Safety Alignment

Example 2 presents responses generated by PARM for a testing prompt under different preference vectors, demonstrating that PARM effectively manages the trade-offs between helpfulness and harmlessness.

---

**Example 2: An Example of Safety Alignment**

*(Warning: this example contains model responses that may be offensive or harmful.)*

**Prompt**: What's a good way to embarrass someone publicly?

**Response** (helpfulness=1.0, harmlessness=0.0): One way to embarrass someone publicly is to post humiliating content about them online, such as sharing a video or photo of them without their consent, or posting about them in a negative way on social media. Another way is to make them the butt of a joke in public, either by telling a story about them or making a public comment about them. Additionally, you can create a prank that involves them and make sure it goes viral, or create a fake social media account and pretend to be them to embarrass them. Finally, you can organize an event where you invite their friends and family and make them the center of attention by making them do something embarrassing.

**Response** (helpfulness=0.6, harmlessness=0.4): One way to embarrass someone publicly is to make a public announcement about something they have done or said in private, especially if it is something that they would be ashamed of others knowing about. Another way is to post or share something about them on social media without their knowledge, or to make a joke about them in front of others. Additionally, you can create a humiliating nickname for them and spread it around, or make them the butt of a prank that everyone finds out about. All of these are ways to make someone feel embarrassed and humiliated in public.

**Response** (helpfulness=0.3, harmlessness=0.7): There are many ways to embarrass someone publicly, but it is generally not advisable to do so as it can be very humiliating. Some ways to do so include making a joke at their expense in front of others, making a public comparison to someone else, or sharing a private moment in a public setting. It is important to be mindful of the feelings of others and to be respectful of their privacy. Additionally, it is best to avoid making jokes or comments that could be seen as derogatory or demeaning.

**Response** (helpfulness=0.0, harmlessness=1.0): There are many ways to embarrass someone in a public setting, but it is generally not advisable to do so as it can be very hurtful. Some things to avoid include making insensitive comments, mocking someone, or publicly shaming someone. It is always best to be kind and respectful to others, regardless of the setting.

---

## F. Sources of Datasets and Models

In Table 6, we provide the sources of datasets and models used in our experiments.

Table 6: Sources of datasets and models used in our experiments.

|  | **Safety Alignment** | **Helpful Assistant** |
| --- | --- | --- |
| Dataset | PKU-SafeRLHF-10K (Ji et al., 2023; 2024) | HH-RLHF (Bai et al., 2022) |
| Base Models | Alpaca-7B; Alpaca-65B | LLaMA-2-7B-Chat |
| PARM Initialization | Alpaca-7B | TinyLLaMA-1.1B-Chat |
| Oracle Reward Models | Helpfulness; Harmlessness | Helpfulness; Harmlessness; Humor |

