# OpenReview forum: "PARM: Multi-Objective Test-Time Alignment via Preference-Aware Autoregressive Reward Model"
_ICML.cc/2025/Conference — ICML 2025 poster_

### Official Review · Reviewer_TCL8 · 2025-03-03

**Overall Recommendation:** 3

**Summary:**

This paper studies the reward-guided multi-objective alignment problem. A prior work GenARM uses a token-level reward model to guide the decoding process, and requires to train two separate reward models to guide the multi-objective decoding process. This work equips GenARM with a preference-aware LoRA-like adapter ($BW_1A+BW_2(\alpha)A$). Their performance is better and more efficient than GenARM.

**Claims And Evidence:**

They claim that PARM is more effective and inference-efficient than GenARM. And this is correct both intuitively and empirically.

**Essential References Not Discussed:**

I don't think there is any work essentially related to the work not cited.

**Experimental Designs Or Analyses:**

The GenARM is the only baseline in evaluation. As shown in Figure 6 of [1], GenARM is not a strong approach for multi-objective alignment (only slightly stonger than RS, which has been beaten by many recent works like [2,3,4]). Therefore, the value of PARM is not very convincing. Since training a DPO model is not harder than training a reward model, it would also beneficial to include comparision with policy-guided approaches.

[1] GenARM: Reward Guided Generation with Autoregressive Reward Model for Test-time Alignment. ICLR 2025.

[2] Decoding-Time Language Model Alignment with Multiple Objectives. NeurIPS 2024.

[3] PAD: Personalized Alignment at Decoding-time. ICLR 2025.

[4] Conditional Language Policy: A General Framework for Steerable Multi-Objective Finetuning. EMNLP finding 2024.

**Methods And Evaluation Criteria:**

The benchmarks (Helpful Assisitant & Safe RLHF) are commonly used in this area. The method PARM makes sense.

**Other Comments Or Suggestions:**

> However, it focuses on aligning with personalized preferences rather than managing trade-offs across different preference dimensions.

I don't think there is much difference between personalization and managing trade-offs, since you can always compare them on a same benchmark.

**Other Strengths And Weaknesses:**

**Strengths**
- This paper is clear and well-written.
- The improvement over GenARM shown in Figure 2 is impressive.

**Weakness**
- The only originality comes from the preference-aware adapter, however, the necessity of this design is not well demonstrated. For example, in Figure 3, SVD-LoRA is comparable with PBLoRA.
- Comparison with policy-guided approaches is missed. And thus it is unknown whether people should use PARM instead of [2,3,4].

**Questions For Authors:**

- In Figure 2(b), why would PARM be much better than GenARM in solely optimizing humor, while only comparable in optimizing helpfulness?
- Why does PARM show better numerical performance than GenARM in Table 1,4, while only slightly better than GenARM in Figure 1,4?

**Relation To Broader Scientific Literature:**

The key contribution of this paper is to propose a preference-aware adapter, which would be of great value if the empirical advantages can be well supported.

The proposed guided-generation process, and the weak-to-strong guidance are already explored by prior works [1,2,3].

**Theoretical Claims:**

The only theoretical claim is Theorem 4.1. I've checked its proof, and it is correct.

---

> ### Author Rebuttal · Authors · 2025-04-01
>
> Thanks for your thoughtful review and valuable feedback. We address your concerns as follows.
>
> > **[Experimental Designs Or Analyses]**. GenARM is the only baseline in evaluation ... include comparision with policy-guided approaches [2,3,4,5].
> > [5] Rewarded soups (NeurIPS 2023)
> > **[Weaknesses 2]**. Comparison with policy-guided approaches.
> > **[Other Comments Or Suggestions]**. difference between personalization and managing trade-offs.
> > **[Relation To Broader Scientific Literature]**. (2) guided-generation process and weak-to-strong guidance are already explored in [1,2,3].
>
> The goal of our PARM is fundamentally different from [2,3,4,5]. Here are the key distinctions (summarized in Table R1 at https://anonymous.4open.science/r/4525-TCL8/rebuttal_to_TCL8.pdf):
>
> - PAD [3] only aligns personalized preferences but **fails to handle trade-offs among multiple dimensions**. Specifically, PAD can handle discrete preference combinations ("helpful", ..., "helpful and harmless", ...) but fails to handle continuous preference combinations (like "60% helpful and 40% harmless"). This limitation is also acknowledged in PAD's rebuttal (see their reply to Weaknesses 1 raised by Reviewer NqpU, https://openreview.net/forum?id=e7AUJpP8bV).
> - MOD [2], CLP [4], and RS [5] **require training $k$ policy LLMs** ($k$ is the number of preference dimensions). This is computationally infeasible, especially when the policy LLM is large (e.g., 65B). Additionally, **MOD** [2] directly combines the logits from multiple trained policy LLMs at inference, causing significant computational overhead, and it **does not contain a guided-generation process and weak-to-strong guidance**. In contrast, **we keep the policy LLM frozen** and use a smaller reward model to guide the larger frozen policy LLM (e.g., 7B guides 65B in our experiment). This avoids training multiple large 65B models as in [2,4,5] and only requires training a 7B reward model, making our method more efficient and promising for users with limited computational resources.
>
> As suggested, we conduct additional experiments to compare PARM with MOD [2] and RS [5]. The experimental setup is the same as in Section 5.1. Results in Figure R1 and Table R2 (in the anonymous link above) show that PARM achieves better Pareto front than RS and MOD, and significantly outperforms them in terms of HV and MIP, demonstrating its effectiveness and high alignment quality.
>
> > **[Relation To Broader Scientific Literature]**. (1) ... preference-aware adapter ... would be of great value if the empirical advantages can be well supported.
> > **[Weaknesses 1]**. the necessity of preference-aware adapter design is not well demonstrated. ... SVD-LoRA is comparable with PBLoRA.
>
> We appreciate your recognition of the originality of our proposed preference-aware adapter (PBLoRA), but we argue that we've provided sufficient evidence for its design necessity in our paper. We clarify the evidence for the necessity of designing PBLoRA as follows.
>
> - **PBLoRA outperforms SVD-LoRA significantly**: (1) The Pareto front of PBLoRA entirely covers that of SVD-LoRA (yellow vs. purple curves in Figure 3). (2) PBLoRA achieves 11.4% and 59.9% improvements in HV and MIP compared to SVD-LoRA (Table 3).
> - **General framework**: As detailed in Lines 220-226 of our paper, SVD-LoRA is a special case of PBLoRA, which means PBLoRA has a greater exploration space and can achieve better results.
> - **Ablation evidence**: In Section 5.3, we systematically analyze different configurations of PBLoRA, showing the effectiveness of each component.
> - PARM with a single PBLoRA **addresses the inefficiency and misalignment issues in GenARM** using $k$ LoRAs. Please refer to our reply to Weaknesses 1 raised by Reviewer 3zGf for details.
>
> > **[Questions 1]**. In Figure 2(b), why would PARM be much better than GenARM in solely optimizing humor, while only comparable in optimizing helpfulness?
>
> Humor is a narrower objective than helpfulness. PARM learns from multiple preference dimensions jointly, benefiting from shared knowledge, while GenARM trains separate reward models for each preference, preventing it from leveraging others data to enhance humor. This difference leads to PARM’s superior performance on the Humor dimension. Conversely, helpfulness, being a broader dimension, can be effectively learned even with GenARM’s isolated approach, resulting in comparable performance on this dimension for both models.
>
> > **[Questions 2]**. Why does PARM show better numerical performance than GenARM in Table 1,4, while only slightly better than GenARM in Figure 1,4?
>
> HV measures the area under the Pareto front. In the Figures, it is clear that PARM has a larger area than GenARM, leading to a higher HV.
>
> MIP measures the uniformity of the solutions on the Pareto front. Obviously, the distribution (the markers in the figures) of PARM is more uniform, so its MIP is higher.

---

> > ### Comment · Reviewer_TCL8 · 2025-04-02
> >
> > > The goal of our PARM is fundamentally different from [2,3,4,5].
> >
> > Thank you for correcting me! Yes, the goal of PAD is indeed different from PARM. But it seems that the goals of MOD [2], CLP [4], and RS [5] are same as PARM, since they are all focusing on balancing different objectives given a human preference vector. I appreciate the supplementary experiments, but I have additional questions:
> > - The MOD, RS approaches can also use low-rank adapters. Considering the fact that PARM uses PBLoRA, it would be unfair to say that MOD and RS are computationally infeasible.
> > - Besides, let's observe the equation (3), which is the same as training a DPO model (just removing the $\pi_\textup{ref}$ model). Thus guiding the generation using token-wise reward model $\pi_\theta$ is equivalent to guiding the genration using DPO model $\pi_\phi/\pi_\textup{ref}$, which has already been well-explored. Therefore, there is not much difference between PARM and policy-guided approaches. If the authors would like to highlight the experimental advantages of token-wise reward models, it would be necessary to polish the story-telling and show more empirical evidence.
> > - As for weak-to-strong guidance, please see Figure 6 in GenARM and Appendix C.3 "Multi-objective proxy-tuning" in MOD. The multi-objective weak-to-strong guidance is not a novel extension. Anyway, not being novel is still acceptable in ICML. This is not a very big issue.
> >
> > And in figure 4, why do the points of GenARM concentrate on helpfulness?
> >
> > ---
> >
> > Update:
> >
> > Q1. MOD can also use 7B model to guide 65B model.
> >
> > I still think the experimental results (only comparing GenARM) are limited.
> >
> > ---
> >
> > Thank you for your updates. Now I can raise my rating to 3, and I hope the authors can put all the contents covered in rebuttal in their submisson later.

---

> > > ### Author Response · Authors · 2025-04-03
> > >
> > > Thanks for your further comments. We deeply appreciate that our previous reply has addressed most of the concerns raised in your initial review. We address the remaining concerns as follows.
> > >
> > > > **Q1**.  the goals of MOD [2], CLP [4], and RS [5] are same as PARM, ... all focusing on balancing different objectives ...
> > > > MOD, RS can also use low-rank adapters. Considering the fact that PARM uses PBLoRA, it would be unfair to say that MOD and RS are computationally infeasible.
> > >
> > > Although MOD, CLP, RS, and our method all target at multi-objective preference alignment, our method based on test-time alignment aims to achieve this with **limited compute resources**.
> > >
> > > To guide a 65B LLM with two preferences (the experiment introduced in Lines 631-652 of our paper), **MOD and RS need to finetune two 65B LLMs**; in contrast, **PARM only needs to finetune a 7B LLM** while keeping the 65B LLM frozen. Obviously, LoRA finetuning a 65B LLM is much more expensive than LoRA finetuning a 7B LLM in terms of computation cost and hardware requirements. For this experiment, our method using PBLoRA can run on one A100 (80G) GPU within 0.85 hours. However, for MOD and RS, finetuning each 65B LLM using LoRA needs 8 A100 (80G) GPUs for 1.65 hours. In total, MOD and RS require $1.65\times 8 \times 2 = 26.4$ GPU hours. Hence, our method is **more memory-efficient and $31\times$ more computationally efficient** than MOD and RS.
> > >
> > > > **Q2**. ... there is not much difference between PARM and policy-guided approaches ...
> > >
> > > Our PARM significantly differs from policy-guided approaches like MOD from the methodological perspective. MOD **independently** trains policy models for each preference dimension **without awareness of other dimensions**, while our PARM trains a **unified** model conditioning on all preference dimensions to **explicitly manage trade-offs between different preferences** (for a preference vector $\alpha$, our model is $\pi_{\theta(\alpha)}$, and the training loss is $\sum_{i=1}^k \alpha_i\ell(\pi_{\theta(\alpha)}, D_i)$).
> > >
> > > Due to independent training, MOD suffers from preference conflicts when combining the logits from multiple policy models during inference; However, our PARM model, thanks to unified training on all preferences, can mitigate preference conflicts and achieve better alignment with preference vectors, as shown by the significantly higher HV and MIP scores (26% and 20% improvements over MOD) in Table R2 in the anonymous link provided in our previous reply (https://anonymous.4open.science/r/4525-TCL8/rebuttal_to_TCL8.pdf).
> > >
> > > A key insight of our paper for the community is that, for multi-objective test-time alignment, our PARM, which trains a **unified** reward model conditioned on **all** preference dimensions, aligns better than GenARM/MOD which combines **separate** reward models trained **individually** for each dimension.
> > >
> > > > **Q3**. ... see Figure 6 in GenARM and Appendix C.3 "Multi-objective proxy-tuning" in MOD. The multi-objective weak-to-strong guidance is not a novel extension.
> > >
> > > Weak-to-strong guidance appears in GenARM, "Multi-objective proxy-tuning" in MOD, and our method. We just want to highlight that our novel method achieves better multi-objective weak-to-strong guidance than GenARM. Specifically, in "1.1B guides 7B" experiment (Table 2), our method achieves 85% and 53% improvements in HV and MIP scores over GenARM. Additionally, in "7B guides 65B" experiment (Table 4), our method shows a 91% improvement in MIP compared to GenARM.
> > >
> > > > **Q4**. in figure 4, why do the points of GenARM concentrate on helpfulness?
> > >
> > > GenARM independently trains reward models for each preference dimension without awareness of each other, resulting in imprecise control over the two competing preferences (helpfulness vs. harmlessness).
> > > To mitigate this issue, our PARM trains a single reward model conditioned on all preference dimensions, leading to better alignment with two preferences.
> > >
> > > ----
> > > > **[Update Q1]**. ... experimental results (only comparing GenARM) are limited.
> > >
> > > As suggested, we conducted an additional experiment to compare PARM with MOD-w2s (the weak-to-strong extension of MOD in Appendix C.3 of the MOD paper) on the helpful assistant task. The experimental setup is the same as in Section 5.2.
> > >
> > > Results are attached at https://anonymous.4open.science/r/4525-TCL8/rebuttal_to_TCL8_2.pdf. Figure R2 shows that PARM has a better and more uniformly distributed front than MOD-w2s and GenARM. Moreover, Table R3 shows that PARM outperforms MOD-w2s (91% improvement in HV, 54% improvement in MIP, and 53% speed-up), demonstrating that PARM is more effective and efficient in multi-objective weak-to-strong guidance.
> > >
> > > ---
> > > ## **Update**
> > > Results of "7B guides 65B"  are attached at https://anonymous.4open.science/r/4525-TCL8/3.pdf, demonstrating that PARM outperforms MOD-w2s in multi-objective weak-to-strong guidance again.
> > >
> > > Thank you for raising the score! We will add all experiments and discussions to our revision.

---

### Official Review · Reviewer_isvK · 2025-03-23

**Overall Recommendation:** 3

**Summary:**

This paper introduces Preference-aware ARM (PARM), a method for guiding large language models (LLMs) at test time based on user preferences. PARM builds upon GenARM, which trains a separate preference model for each human preference. In contrast, PARM employs a unified model that conditions all preferences on a single vector, enabling more flexible adaptation. Compared to GenARM, PARM generates responses that better align with human preferences.

**Claims And Evidence:**

The experimental results support the paper’s claim that PARM enhances response alignment with human preferences. However, my main concern is the significance of the research problem. Controlling text generation through prompt engineering is a straightforward alternative that can improve alignment with human preferences without requiring an additional model for preference guidance. This approach is not considered a baseline in the paper, which raises questions about the necessity of training an extra model. Additionally, relying on an additional 7B or smaller LLMs for preference guidance may not be a practical solution due to the computational overhead.

**Essential References Not Discussed:**

The paper overlooks several key works on controllable text generation at inference time. For instance, "Controllable Text Generation for Large Language Models: A Survey" provides a comprehensive overview of various methods for controlling text generation. Additionally, "Plug and Play Language Models: A Simple Approach to Controlled Text Generation" is one of the pioneering works in this area, introducing a flexible approach to guiding language models without retraining. Including discussions on these references would strengthen the paper’s positioning within the broader literature.

**Experimental Designs Or Analyses:**

I examined both the qualitative and quantitative analyses, and they appear well-structured and sound.

**Methods And Evaluation Criteria:**

The methods and evaluation metrics make sense to me.

**Other Comments Or Suggestions:**

Please refer to Strengths And Weaknesses.

**Other Strengths And Weaknesses:**

Strengths:
1. The paper is well-structured and easy to follow.
2. The proposed method is clearly explained and logically sound.
3. The experimental results effectively demonstrate the method’s effectiveness in improving alignment with human preferences.

Weaknesses:
1. The significance of the research problem is unclear, as human preference alignment in LLM generation can often be achieved through prompt engineering without additional models.
2. The approach introduces additional inference costs and requires extra training data, which may make it impractical compared to simpler alternatives.

**Questions For Authors:**

Can a much smaller language model (e.g., less than 1B parameters) achieve comparable performance while significantly reducing inference latency?

**Relation To Broader Scientific Literature:**

Controllable text generation for LLMs in inference time

**Theoretical Claims:**

I quickly went through the Theorems.

---

> ### Author Rebuttal · Authors · 2025-04-01
>
> Thanks for your thoughtful review and valuable feedback. We address your concerns as follows.
>
> > **[Essential References Not Discussed]**. Discussions on controllable text generation are missing.
>
> Controllable Text Generation (CTG) generates text from LLMs with specific attributes or constraints. Our PARM is a specific CTG implementation for multi-objective test-time alignment. Compared to the mentioned method PPLM (ICLR 2020), which uses multiple attribute models and requires forward and backward passes during generation, PARM employs a single reward model to dynamically adjust text during inference, achieving lower computational costs.
>
> We will expand this discussion and include the CTG references in the revision.
>
> > **[Weaknesses 1]**. The significance of the research problem is unclear, as human preference alignment in LLM generation can often be achieved through prompt engineering without additional models.
> > **[Claims And Evidence]**. prompt engineering is not considered a baseline.
>
> **Prompt engineering alone is insufficient for aligning LLMs with complex human preferences**. Most successful alignment methods (like RLHF) require post-training, and our method can be viewed as a test-time alternative to post-training to reduce training computations.
>
> As suggested, we conducted an additional experiment to compare our method with a prompt-based baseline whose instruction is "Please ensure your response is X" (adapted from Personalized Soups (NeurIPS 2024 workshop) and PAD (ICLR 2025)), where "X" is "helpful", "harmless", or "a% helpful and b% harmless".  The experimental setup is the same as in Section 5.1.
>
> As shown in https://anonymous.4open.science/r/4525-isvK/rebuttal_to_isvK.pdf, prompting has little effect on both two preferences. Moreover, **prompting cannot achieve precise control over preferences trade-offs** and its results fail to form a Pareto front,  demonstrating that **prompting is not a good choice for aligning LLMs with complex human preferences**. In contrast, our method has a much better Pareto front, enabling precise control over multiple competing preferences simultaneously.
>
> > **[Weaknesses 2]**. introduces additional inference costs and requires extra training data ... impractical compared to simpler alternatives.
> > **[Claims And Evidence]**. relying on an additional 7B or smaller LLMs for preference guidance ... computational overhead.
>
> As discussed in our reply to Weaknesses 1, the prompt-based method, a simpler alternative, is not effective enough to align LLMs with complex human preferences.
> Recently, many advanced alignment methods (e.g., RLHF, GenARM) have emerged, and our work extends them to multi-objective test-time alignment.
>
> **Training data problem**: we would like to clarify that obtaining a multi-objective preference dataset is **not very difficult** and can be derived from a traditional single-objective dataset (see our reply to Weaknesses 2 raised by Reviewer 3zGf).
>
> **Inference cost problem**: we would like to clarify that the increase in inference cost is **small** and can be effectively mitigated through distributed deployment as follows:
> - **Addressed by distributed deployment**. Our method uses a single reward model to guide the frozen policy LLM. Since both models can generate the next token in parallel, the inference time remains the same as using the frozen LLM directly.
> - **Without distributed deployment, our method is still practical** for two reasons: (i) our method enables weak-to-strong guidance, such as using a 7B reward model to guide a frozen 65B LLM in our experiments. It increases inference time by about 30% compared to directly using the 65B LLM. We believe that the increase in inference time is worthwhile since our method **significantly improves preference alignment without extensive training of the large policy LLM** (Figure 4). (ii) Compared with GenARM, our method significantly reduces inference costs. Please refer to our reply to Weaknesses 1 raised by Reviewer 3zGf for details.
>
> Hence, by leveraging a smaller reward model to guide a larger LLM, PARM offers a practical and efficient method for achieving multi-objective alignment without requiring extensive training. This is especially beneficial for users lacking resources to fine-tune the policy LLM.
>
> > **[Questions]**. Can a much smaller language model achieve comparable performance while significantly reducing inference latency?
>
> There is a trade-off between the capacity of the reward model and its guiding effect. An extremely smaller reward model can reduce inference costs but may also compromise the guiding effect. Note that in our experiments, the reward models are already very small compared to the frozen LLMs (e.g., 7B vs. 65B and 1.1B vs. 7B). Moreover, as discussed in our reply to Weaknesses 2, the inference cost problem can be effectively mitigated through distributed deployment. Thus, we leave the exploration of much smaller reward models for future work.

---

### Official Review · Reviewer_3zGf · 2025-03-24

**Overall Recommendation:** 3

**Summary:**

The authors proposed a preference-aware ARM for multi-objective test-time alignment. PARM is an ARM conditioned on user preferences through the proposed PBLoRA, which manages trade-offs across multiple preference dimensions during inference.

**Claims And Evidence:**

Yes.

**Essential References Not Discussed:**

NA.

**Experimental Designs Or Analyses:**

Yes.

**Methods And Evaluation Criteria:**

Yes.

**Other Comments Or Suggestions:**

NA.

**Other Strengths And Weaknesses:**

Strengths of the paper:
1. The paper is well-written and easy to follow.
2. The problem is of great value to investigate.
3. Source code of the proposed model is provided in the paper.

Weaknesses of the paper:
1. The proposed model is the extension of the existing model ARM and in particular the GenARM. The main difference is that the proposed model is to condition the massive of model parameters on the k-dimensional preference vector alpha. Given this, the contribution of the paper sounds limited.
2. As the proposed model works for multi-objectives, training material needs to be sufficient for training the model, which may be bottleneck in some setting where training data are difficult to obtain.
3. To achieve the goals of multi-objectives learning, there are a number of strategies. And one of them is the studied strategy in this paper, i.e., introducing an autoregressive reward model. Why don’t the authors consider fine-tune the original model?

**Questions For Authors:**

Please see the above weaknesses.

**Relation To Broader Scientific Literature:**

The problem studied in the paper is interesting to many researchers.

**Theoretical Claims:**

Yes.

---

> ### Author Rebuttal · Authors · 2025-04-01
>
> Thanks for your thoughtful review and valuable feedback. We address your concerns as follows.
>
> > **[Weaknesses 1]**. The proposed model is the extension of the existing model ARM and in particular the GenARM. The main difference is that the proposed model is to condition the massive of model parameters on the k-dimensional preference vector alpha. Given this, the contribution of the paper sounds limited.
>
> While PARM builds upon the foundation of ARM and GenARM, it introduces significant innovations and improvements that address key limitations of GenARM. We clarify it in detail as follows.
>
> Unlike GenARM, which requires multiple ARMs for different preference dimensions, PARM uses a single ARM conditioned on the preference vector, resulting in several advantages:
> - **More Parameter-efficiency**: GenARM requires storing $k$ separate ARMs, while PARM needs only a single one, making it approximately $k\times$ more parameter-efficient (Table 2 in our paper).
> - **Faster Inference**: PARM is significantly faster during inference (Table 2), as it computes rewards from a single ARM rather than $k$ separate ARMs in GenARM.
> - **Better control of trade-offs with preference vector**: In GenARM, the $k$ ARMs are trained independently on different preference dimensions without awareness of each other, leading to potential conflicts when their rewards are combined during inference. PARM, on the other hand, is explicitly trained to manage trade-offs between different preferences (as detailed in Section 4.3).  As a result, our model's behavior aligns more directly with the specified preference vector, making control more intuitive and predictable. This leads to better alignment with user-specified preferences and simplifies usage, as demonstrated by the significantly higher HV and MIP scores in Tables 1 and 2. Examples 1 and 2 in our paper further demonstrate how PARM effectively balances competing objectives like helpfulness and harmlessness according to the specified preference weights.
>
> Therefore, our PARM provides a more efficient, scalable, and controllable method for multi-objective test-time alignment of LLMs. We believe these contributions are substantial and address important challenges in the field of preference-aligned language models.
>
> > **[Weaknesses 2]**. As the proposed model works for multi-objectives, training material needs to be sufficient for training the model, which may be bottleneck in some setting where training data are difficult to obtain.
>
> We appreciate your concerns regarding the potential bottleneck of training data for multi-objective models. We want to point out that **multi-objective dataset is not very difficult to obtain, as it could be derived from a traditional single-objective dataset as follows**.
>
> Specifically, we can take standard preference datasets $\\{x, y^1, y^2, z\\}$ (which are widely available, $y^1$ and $y^2$ are two different responses to the prompt $x$, and $z=1$ if $y^1$ is better than $y^2$, otherwise $0$) and extend them to multi-objective datasets $\\{x, y^1, y^2, z_1, \dots, z_k\\}$ by adding preference labels $z_i$ for different dimensions. These additional labels can be obtained using GPT judges or publicly available reward models/classifiers. For example, in our Helpful Assistant experiment (Section 5.2), the humor preference labels were obtained using a public classifier (as detailed in footnote 10), following the approach of previous works [1,2].
>
> [1] MetaAligner: Towards generalizable multiobjective alignment of language models. NeurIPS 2024.
>
> [2] Rewards-in-context: Multi-objective alignment of foundation models with dynamic preference adjustment. ICML 2024.
>
> > **[Weaknesses 3]**. To achieve the goals of multi-objectives learning, there are a number of strategies. And one of them is the studied strategy in this paper, i.e., introducing an autoregressive reward model. Why don’t the authors consider fine-tune the original model?
>
> While **fine-tuning** the original model is a possible approach, this method **requires enormous computational resources** that many researchers and practitioners cannot access when the model is **large** (such as Alpaca-65B).
>
> Our PARM is a **test-time alignment** approach that addresses this limitation by training a **small** reward model rather than the original LLM, reducing computation cost largely and **making multi-objective alignment accessible with limited computing resources**.
>
> Our method enables **weak-to-strong guidance, allowing a smaller reward model to guide a larger frozen LLM without expensive training**. For example, in our experiments, a 1.1B reward model guides a 7B LLM, and a 7B reward model guides a 65B LLM. This capability is particularly valuable for users who cannot afford to train large LLMs but still need to leverage their capabilities.
>
> Therefore, based on test-time alignment, our PARM provides **a practical solution** that makes multi-objective alignment more accessible to the broader research community.

---

### Decision · Program_Chairs · 2025-05-01

**Decision:**

Accept (poster)

**Comment:**

This work proposes a multi-objective test-time alignment method, named PARM, that improves on the previously proposed GenARM technique by training a single "unified" Autoregressive RM in terms of multiple ones, which is shown to outperform GenARM as well as other baselines in the empirical validation of the method.

Although reviewers initially had some concerns, in particular regarding novelty and lack of comparison to some other baselines, the authors did a thorough job to try and address them, eventually convincing all reviewers to advocate for acceptance. I believe this is indeed interesting work that would deserve to be accepted, though I also share the concern that this remains somewhat incremental work over GenARM, with a relatively "niche" application domain (test-time alignment for multi-objective preferences). I am thus recommending a "Weak accept", similar to all three reviewers.